# Perceived overqualification as a double-edged sword for employee creativity: The mediating role of job crafting and work withdrawal behavior

**Daokui Jiang** [ORCID]*, **Lei Ning, Yiting Zhang**

Business School, Shandong Normal University, Jinan, China

* jdk240504@163.com

## Abstract

With the continuous development of education level and the downturn of economic situation, employment competition is intensifying, more and more high-quality talents appear, and the misfit between people and posts has become a common phenomenon. However, there is no consensus on the relationship between perceived overqualification and employee creativity. Based on the conservation of resource theory, this study reveals the micro mechanism and boundary conditions of the influence of excessive qualification on employee creativity. This study analyzed 487 valid samples obtained in three stages. The results show that: (1) Job crafting has a positive mediating effect on perceived overqualification and creativity, and the path of the two halves is positive; (2) Work withdrawal behavior plays a negative mediating role between the perceived overqualification and creativity. The path in the first half is positive, and the path in the second half is negative; (3) Organizational identity moderates the effect of perceived overqualification on job crafting and work withdrawal behavior. Specifically, the higher the sense of organizational identification, the stronger the positive effect of perceived overqualification on job crafting and the weaker the positive effect on work withdrawal behavior; (4) Organizational identification moderates the mediating role of job crafting and work withdrawal behavior in the relationship between overqualification and creativity. Specifically, the higher the organizational identity, the stronger the indirect positive effect of perceived overqualification on creativity through job crafting, and the weaker the indirect negative impact of perceived overqualification on creativity through work withdrawal behavior. The study conclusion deepens the research on the mechanism of the influence of the perceived overqualification on employees' work behavior, and provides practical enlightenment for the organization and management of employees with excess qualification.

## Introduction

Data from the seventh national census shows that the education level of China's population has improved significantly. In the past 10 years, the average number of years of education for

**Funding:** The author(s) received no specific funding for this work.

**Competing interests:** The authors have declared that no competing interests exist.

the working-age population aged 16–59 has increased from 9.67 years in 2010 to 10.75 years, indicating that the quality of China's population pouring into social employment is constantly improving. However, with the increasing scale of supply of high-quality talents and the saturation of post demand, the phenomenon of mismatch between talents and position is becoming more and more common. Especially after the outbreak of the COVID-19, the labor market has become more competitive, and more people choose to take jobs that are below their education level, skills, experience, and other qualities. Due to many social factors such as education, economy and competition for employment, more and more people are in an underemployment situation and a large number of employees say that they are experiencing overqualification [1], i.e., the qualifications of individuals such as education level, work experience and skills exceed the job demands [2]. Perceived overqualification refers to the employees' subjective perception that their own abilities exceed the job demands, which is a subjective feeling that can affect the employees' work attitudes, behavior and performance, etc. Most of the existing studies on perceived overqualification focus on the negative side, stating that perceived overqualification negatively predicts organizational commitment, job satisfaction, and job engagement [3] as well as positively predicts the turnover tendency [4], job boredom [5], and counterproductive work behavior [6]. As the research keeps going on, some scholars also focus on the positive effects of perceived overqualification, pointing out that perceived overqualification positively affects organizational citizenship behavior [7], employee creativity [8], and deviant innovation [9]. In the current VUCA (Volatility, Uncertainty, Complexity and Ambiguity) era, companies are increasingly dependent on talented people for innovation, so exploring the relationship between perceived overqualification and employee creativity has become a key focus and hot topic of research in both theoretical and practical circles.

In fact, scholars have not yet agreed on the relationship between perceived overqualification and employee creativity. There are two types of representative views in the scholarly community. The first one is based on a competency perspective that views perceived overqualification as an employee advantage. This type of view sees overqualified employees having a higher self-efficacy [8] and being able to master their task easier and faster and then showing higher creativity [10], thus perceived overqualification positively predicts on employee creativity [11]. Another view is based on the perspective of employee motivation. This perspective argues that despite the high competence of overqualified employees, perceived overqualification weakens employees' motivation and thus negatively affects employee creativity [12]. Employees who feel overqualified usually have difficulty in meeting their personal needs to fully develop their abilities and gain higher job status and social recognition from the work they complete in the organization, thus breaking the psychological contract and perception of fairness between the employee and the organization and leading to emotional exhaustion, which negatively affects employee creativity [13].

Both of above perspectives provide useful insights into understanding the relationship between perceived overqualification and employee creativity, but there are few studies that explore the integrated effects of perceived overqualification on employee creativity. According to conservation of resources theory, when employees face the threat of resource loss or when there is an actual loss of resources, two different responses occur [14, 15] - resource investment and resource conservation. Overqualified employees have more personal resources such as knowledge, skills, and experience than the job demands[16]. On the one hand, they tend to pursue resource gain, actively invest resources in their work, and are willing to take advantage of their strengths to improve their work by job crafting behaviors such as increasing the difficulty and complexity of their work content, adding challenging work tasks, and other behaviors, thus stimulating their personal creativity. On the other hand, overqualification puts employees in a state of resource wasting [17], where employees lack the opportunity to utilize

their own redundant resources, resulting in frustration, loss, guilt, etc. This continuous state makes them burn out and depletes a lot of mental resources. In order to avoid further depletion of resources, they tend to conserve resources, for example, by avoiding work tasks and doing other work withdrawal behaviors to cope with work negatively, thus reducing creativity.

This study responds to the call of Simon et al to comprehensively explore the impact of perceived overqualification on employees [18], incorporates both positive and negative effects of perceived overqualification into the study based on conservation of resources theory, and proposes a research model in which perceived overqualification affects employee creativity through the dual paths of job crafting and work withdrawal behavior, respectively. Meanwhile, considering the possible boundary conditions of these two paths, this paper introduces organizational identity as a moderating variable to explore how to strengthen the positive side and weaken the negative side of perceived overqualification under the effect of organizational identification. This study has three main contributions. First, different from previous studies that only focus on the positive or negative effects of perceived overqualification, this paper proposes two distinct effect paths based on conservation of resources theory, which explains the inconsistent findings in the previous literature on "the effects of perceived overqualification on employee creativity"; Secondly, most of the existing studies on perceived overqualification are based on relative deprivation theory [19, 20], this study takes conservation of resources theory as the theoretical basis, which enriches the theoretical perspective in the field of overqualification and extends the theoretical results of conservation of resources theory. Finally, by including the contextual factor (organizational identification) as a moderating variable, the role of employee-organization relationship in the process of perceived overqualification affecting creativity is further clarified, and the research related to organizational identification is enriched.

## Theoretical background and hypotheses

**Conservation of resources theory.** Conservation of resources theory suggests that individuals always tend to maintain, protect, and acquire resources that individuals perceive as valuable or ways to acquire value [21]. Conservation of resources theory has two core principles [22]. The first is the principle of resource investment, whereby individuals must invest resources in order to repair resource losses and acquire new resources. The second is the principle of priority of resource loss, which means that the psychological harm brought to individuals du Resource conservation theory reflects the interaction between resources, and overqualification is essentially a manifestation of employee resource surplus. e to resource loss has a much greater effect than the psychological help brought by resource gain. Conservations of resource theory reflect the interaction between resources, and overqualification is an expression of employee resource surplus in essence. On the one hand, faced with their own abundant resources, overqualified individuals are more likely to engage in resource investment behaviors to obtain more resource. On the other hand, the perception of wasted resources may trigger the individual's resource protection mechanism, and thus stop investing in resources. Thus, the process by which the perception of overqualification affects employee creativity may present two distinct paths of resources investment and resources conservation. Specifically:

1. Path 1: resources investment. Conservation of resources theory suggests that resource investment promotes resource acquisition, creating a resource "value-added spiral" [15]. Individuals who are abundant in resources have more opportunities to acquire new resources through resource investment, while individuals who are lacking in resources are more likely to suffer from resource loss. Individuals who are abundant in resources have more opportunities to acquire new resources through resource investment, while individuals who are lacking in resources are more likely to suffer from resource loss. Overqualified

employees have more resources, and when they see their surplus resources as an advantage and opportunity, they actively seek fulfillment in their work and are more motivated to engage in resource investment behaviors in the hope of gaining resources, thus forming a resource gain spiral and using their surplus qualifications to crafting their surroundings or individual behaviors to promote organizational innovation.

2. Path 2: resources conservation. Conservation of resources theory states that the loss of resources can cause stress to individuals [15]. Perceived overqualification triggers the perception that employees' own resources are wasted, prompting negative emotions such as feelings of relative deprivation and work boredom [23]. Negative emotions consume a lot of individual mental resources, and in order to maintain their own resources, employees often refuse to invest further resources in their work, manifesting themselves in work withdrawal behaviors such as negativity, tardiness and early leaving, which negatively affect their creativity level

In addition, conservation of resources theory suggests that environmental conditions can both nurture and nourish the resources that exist within them, as well as play a limiting and hindering role [22] this implies that the organization in which an individual works plays an important role in shaping and sustaining his or her resources. An individual's identification with the organization influences his or her attitude towards work and performance. Therefore, this paper introduces the variable of organizational identification as a boundary condition for the effect of perceived overqualification on employee creativity to further clarify the mechanism of the effect of overqualification on creativity.

This study examines the effects of perceived overqualification on employee creativity based on conservation of resources theory: (1) to explore the effects of perceived overqualification on employee creativity through the path of "resources investment"; (2) to explore the effects of perceived overqualification on employee creativity through the path of "resources conservation"; and (3) to explore the moderating role of contextual factors (organizational identification) in the above path. By studying the above, this study tries to explore the mechanism of perceived overqualification on employee's creativity.

**Resources investment path: The mediating role of job crafting.**   Overqualification refers to the fact that employees' knowledge, skills, education, and work experience exceed the job requirements [24], which is a job misfit phenomenon. When there is a job misfit, employees usually try to change their jobs to reduce the difference between their reality and their ideal jobs, and this process is called job crafting. Job crafting refers to the behavior of individuals actively adjusting their jobs in order to pursue their goals, such as improving work processes and expanding work boundaries. Research has shown that personal and job mismatch is an important trigger for job crafting [25].

According to conservation of resources theory, individuals with more resources have more opportunities to invest in resources, forming a resource gain spiral. This gain spiral allows employees to feel a positive stimulus. Job reinvention is the behavior of employees who take the initiative to change and design their own work at work, which consumes a lot of individual resources. Overqualified employees have more opportunities to use their surplus skills to proactively reinvent their jobs to create a better work environment [26], thus reducing the difference between reality and their ideal job. Research has shown that the realization of self-worth, the construction of a positive image, and the need for job control are the main motivations for employees to engage in job crafting [27]. Perceived overqualification may stimulate these types of needs of employees and thus motivate individuals to invest resources in their work to craft their work. First, overqualified employees perceive themselves to be outstanding and prefer to do well in the workplace to earn the respect of others [28]; therefore, perceived

overqualification trigger the need for employees to realize their self-worth and construct a positive image [19]. Second, overqualified employees have competencies beyond the job demands, so they are able to perform work tasks more effectively and learn new tasks, technologies, etc [5]. In order to reduce individual resource waste, surplus individual resources stimulate employees' need for job control. Increased job control can help employees gain resource benefits [24], such as employees optimizing their own work processes through careful design to improve efficiency and gain new skills at the same time.

In summary, job crafting is an active bottom-up resource building strategy made by employees on their own initiative. An employee's perceived overqualification means that the employee has more competencies, skills, and experience than he or the job demands. Somehow they are able to do their jobs faster and better and thus overqualified employees have the capacity to actively craft their daily work content and environment to make it more responsive to their needs. Perceived overqualification increases the motivation of employees to pursue resource gain and invest resources to transform their work through job crafting. Based on the above analysis, this paper proposes the hypothesis:

H1: Perceived overqualification has positive effects on job crafting.

Employee creativity refers to the ability of employees to generate novel ideas, services, processes, etc. in the course of their work [29]. According to the job demands-resources model, job crafting behaviors are positive and effective resources-building behaviors that can be divided into four specific strategies: increasing social job resources, increasing structural job resources, increasing challenging job tasks, and decreasing hindering job tasks [30]. Job crafting can motivate employees and promote their creativity, which in turn provides strong support for corporate innovation [31]. First, job crafting behaviors such as increasing structural and social work resources can lead to continuous learning and increased work engagement [25], which in turn leads to higher levels of creativity [32]. Second, increasing challenging work tasks motivates employees to seek new resources, and completing challenging work tasks reduces work boredom due to overqualification and motivates employees, which in turn promotes their creativity performance [33]. Finally, reducing hindering work tasks would reduce job burnout, which could also promote employee creativity. In conclusion, job crafting reflects employees' initiative in their work and helps to enhance creativity. Based on the above analysis, this paper proposes the hypothesis:

H2: Job crafting has positive effects on employee creativity.

Tims and Bakker define job crafting as "the behavioral changes that employees make to balance job demands and job resources according to their own abilities and needs" [30]. Perceived overqualification is triggered by the individual's perception that his or her job resources exceed the job demands [34]. Therefore, if overqualified individuals regard their own resources as a unique advantage and want to gain more resources, they make resource investment in their work to craft their own work and thus enhance their creativity, forming a resource gain spiral. Combining hypothesis H1 and hypothesis H2, perceived overqualification positively affects job crafting, and job crafting positively affects employee creativity. Therefore, it is reasonable to infer that job crafting plays a mediating role in the relationship between perceived overqualification and employee creativity, so this paper proposes the hypothesis:

H3: Job crafting positively mediate the relationship between perceived overqualification and employee creativity.

**Resources conservation path: The mediating role of work withdrawal behavior.** Work withdrawal behavior is a set of negative behaviors that employees adopt to avoid work or psychologically disengage from the organization, which are manifested by being late, leaving early, and not performing at work [35]. Work withdrawal behavior is developed gradually at work [36], starting with deviations in thinking and gradually manifesting later as slowness at work and even a turnover intention. Perceived overqualification leads to a perception of resources being wasted, and based on the principle of priority of resource loss, employees who perceive that their resources are being wasted will commit to maintaining their existing resources, refuse to invest in resources or even show withdrawal behaviors at work [37]. When employees perceive that they are overqualified, the work they perform or the opportunities they obtain in the organization do not meet their needs to fully develop their abilities, obtain higher job status and social recognition, which leads to a sense of person-job misfit and breaks the psychological contract and perception of equity between the employee and the organization, and creates a sense of relative deprivation [38]. These cognitive and psychological states make the sense of overqualification bring about various negative outcomes. These cognitive and psychological states make perceived overqualification bring about various negative outcomes. Due to the inability to fully utilize their personal talents at work, employees experience job boredom and frustration, leading to depletion of individual self-control resources and work withdrawal behaviors. In addition, before investing resources, individuals evaluate the resource benefits they can obtain and the resource losses they may face, and then decide whether to invest resources. According to equity theory, employees compare the resources they receive with what others receive, and overqualified employees compare the benefits they receive with other employees in the same position, and if they find that employees who do not invest as much resources as they do receive the same resources as they do, they will invest less resources [9]. Since work withdrawal behavior does not require investment of time, energy, etc., job withdrawal is, to some extent, a reasonable resource conservation strategy [39]. Therefore, this study argues that perceived overqualification triggers employees' resource conservation mechanisms that lead to their work withdrawal behavior. Based on the above analysis, this paper proposes the hypothesis:

H4: Perceived overqualification has positive effects on work withdrawal behavior.

Although work withdrawal behavior helps employees conserve certain personal resources, it still comes at a cost. Conservation of resources theory suggests that any gain or loss of resources has some impact [14]. Researches have shown that work withdrawal behaviors are negative behaviors in the workplace, and negative behaviors in the workplace depletes individuals' mental resources, leading to self-control resources depletion, which in turn reduces work engagement, creating a vicious cycle [40]. Employees' creativity is manifested in their initiative to change the status and generate new ideas. Whether employees show their creativity at work depends largely on their subjective consciousness, and their creativity will be weakened when they engage in work withdrawal behavior. Work withdrawal behaviors can be specifically divided into psychological withdrawal and behavioral withdrawal [41]. Psychological withdrawal behavior is manifested as psychological separation from work, such as being distracted during work and developing turnover intention, these psychological withdrawal behaviors makes employees barely complete their work tasks instead of being active and creative. Behavioral withdrawal behavior means that employees show negative behaviors in the workplace, such as being late, leaving early and deliberately delaying work progress. Behavioral withdrawal behaviors also mean that employees reduce their physical and mental commitment to their work, while reducing the altruistic behavior of job creation. In general, both psychological withdrawal and behavioral withdrawal deplete employees' mental resources, which

stimulate resource conservation mechanisms and lead to employees reducing their resource input at work and not showing higher levels of creativity at work. Based on the above analysis, this paper proposes the hypothesis:

H5: Work withdrawal behavior has negative effects on employee creativity.

Combined with the above analysis, the pressure of resource loss caused by perceived overqualification triggers a resource conservation mechanism for employees. Work withdrawal behaviors do not require a significant investment of time and energy, etc., and is an effective method of resource conservation. Therefore, perceived overqualification triggers work withdrawal behavior and this negative behavior leads to a decrease in the level of employee creativity. Combining H4 and H5, perceived overqualification positively affects work withdrawal behavior, and work withdrawal behavior negatively affects employee creativity. Therefore, it is reasonable to infer that work withdrawal behavior plays a mediating role in the relationship between perceived overqualification and employee creativity. Based on the above analysis, this paper proposes the hypothesis:

H6: Work withdrawal behavior negatively mediates the relationship between perceived overqualification and employee creativity.

**The moderating role of organizational identification.** Organizational identification refers to the degree to which employees identify with the organization and can influence their attitudes and behaviors toward work. Harari et al found that overqualified employees may have excellent performance but may lack motivation [3]. The employee-organization relationship, as a particular construct of individual cognition and emotion, can provide motivation for positive attitudes and behaviors. It has been noted that employees with high organizational identification are more willing to invest valuable personal resources such as time and energy in their work [42], suggesting that employees with different levels of organizational identification within the same organization may perform differently at work. On the one hand, according to Maslow's hierarchy of needs theory, individuals have the need for belonging and respect, i.e., they want to be connected to others, be respected and cared for, and are of value to others. If employees have a high level of organizational identification, they have more trust in the organization or their colleagues, and at this time they are more willing to use their knowledge and skills, etc. to help their colleagues and thus earn organizational respect [43]. This helps overqualified employees to perceive that they can use their surplus resources to better control their work environment. On the other hand, overqualified employees have the need to utilize their surplus resources. When organizational identity is at a high level, employees have a strong sense of belonging to the organization and show a high level of identification with the organization's culture, goals, and values [44]. At this time, organizational goals and individual goals are aligned. In order to prove their competence and accomplish their goals, employees strive to accomplish their work tasks as well as organizational goals [45]. As a result, overqualified employees with a higher organizational identification are more motivated to put resources to work and more likely to do job crafting behaviors. On the contrary, when organizational identity is low, employees have a low sense of belonging to the organization. At this time, employees' personal goals are not in line with organizational goals, and they may attribute their overqualification to organizational arrangements, creating a feeling of "not having talent" in the organization, which consumes a lot of mental resources. In turn, employees' resource conservation mechanism is stimulated, they refuse to invest personal resources in their work, thus reducing work reinvention behavior. Based on the above analysis, this paper proposes the hypothesis:

H7: Organizational identification moderates the positive relationship between perceived overqualification and job crafting, such that this relationship becomes stronger when organizational identification increases.

The studies found that employees' job withdrawal behavior was closely related to their perceived relationship with the organization. When employees' organizational identification at a high level, they trust the organization more, see themselves as "insiders" of the organization, and consider organizational goals as their own [46]. At this time, employees think highly of their organizations, are happy to work for them. They believe that they can realize their value in the organization, and exhibit positive, confident, and energetic characteristics. This positive state can weaken negative emotions such as relative deprivation triggered by perceived overqualification and reduce employees' work withdrawal behaviors [47]. Conversely, when employees' organizational identification is at a low level, employees are unable to develop a sense of belonging, at which they feel that they are involuntarily accepting overqualified jobs, make negative assessments of their jobs, and believe that their abilities will not be needed by the organization, thus more likely to do work withdrawal behavior [48]. In addition, organizational identification is also reflected in employees' desire to maintain their membership in the organization. Employees with high organizational identification have better loyalty to the organization and do not harm the organization's interests. In this case, employees are more willing to attribute their overqualification to factors such as fierce competition and see it as an advantage that they want to use to contribute to the organization. Conversely, employees with low organizational identification view their membership in the organization as dispensable and are less loyal to the organization. They do not focus on organizational goals and devote more resources to their own interests, thus exhibiting more work withdrawal behaviors. Low organizational identification can reinforce negative perceptions such as relative deprivation brought by perceived overqualification and reduce employees' creativity levels. Based on the above analysis, this paper proposes the hypothesis:

H8: Organizational identification moderates the positive relationship between perceived overqualification and work withdrawal behavior, such that this relationship becomes weaker when organizational identification increases.

As already mentioned in H3 and H7, we hypothesize that organizational identification moderate the effects of perceived overqualification on job crafting, which will in turn influence creativity. Specifically, high organizational identification stimulates employees' work motivation and offers them organizational and emotional resources, making them more likely to put their surplus resources to work. By focusing on the positive aspects of overqualification, employees' creativity will be enhanced. Therefore, this paper proposes the hypothesis:

H9: Organizational identification positively moderates the indirect effect of perceived overqualification on employee creativity through job crafting, such that this indirect effect will be stronger when organizational identification is higher rather than lower.

As already mentioned in H4 and H8, we hypothesize that organizational identification moderate the effects of perceived overqualification on work withdrawal behavior, which will in turn influence creativity. Specifically, employees with low organizational identity are less loyal to the organization and more likely to attribute overqualification to the organization, making them less likely to put their surplus resources to work.By focusing on the negative aspects of overqualification, employees' creativity will be weakened. Therefore, this paper proposes the hypothesis:

H10: Organizational identification negatively moderates the indirect effect of perceived overqualification on employee creativity through work withdrawal behavior, such that this indirect effect will be weaker when organizational identification is higher rather than lower.

In sum, we propose the model in Fig 1, in which creative performance pressure relates positively to job crafting and work withdrawal behavior, which subsequently relate to employee creativity. Further, the indirect effects are moderated by organizational identification.

## Methods

### Sample and procedure

The study does not involve animal subjects. The study does not involve human subjects. This manuscript does not include any potentially identifiable human images/data, including individual case descriptions. The research was conducted in five companies in eastern China, and the process was as follows: First, after seeking permission from company leaders, the researchers contacted the department heads to introduce the purpose of the study and explain the potential benefits of the study. An electronic questionnaire was also sent to the department heads for feedback, and the questionnaire was revised in detail based on the feedback. Next, researchers sent emails to employees who were eligible for the study explaining the study and its purpose, and obtained voluntary participation from 556 employees. Finally, the researchers communicated with the finalized subjects to explain the procedures of the study, emphasizing that the study was anonymous and the questionnaire data were kept strictly confidential. Finally, an electronic questionnaire was sent to the employees who volunteered to participate in this study according to the procedure, and each participant was paid a certain amount of money at the end of the study.

To reduce common method bias, the research data were collected at three time points, considering that if the time interval was too long, it might obscure the existing relationships among the variables, and if the time interval was too short, it might exaggerate the relationships among the variables. Therefore, each phase of this research was spaced 3 weeks apart. Prior to the start of the research process, the researchers coded the employees who participated in the study for matching at the end of the survey. In Phase I (T1: June 2022), employees report

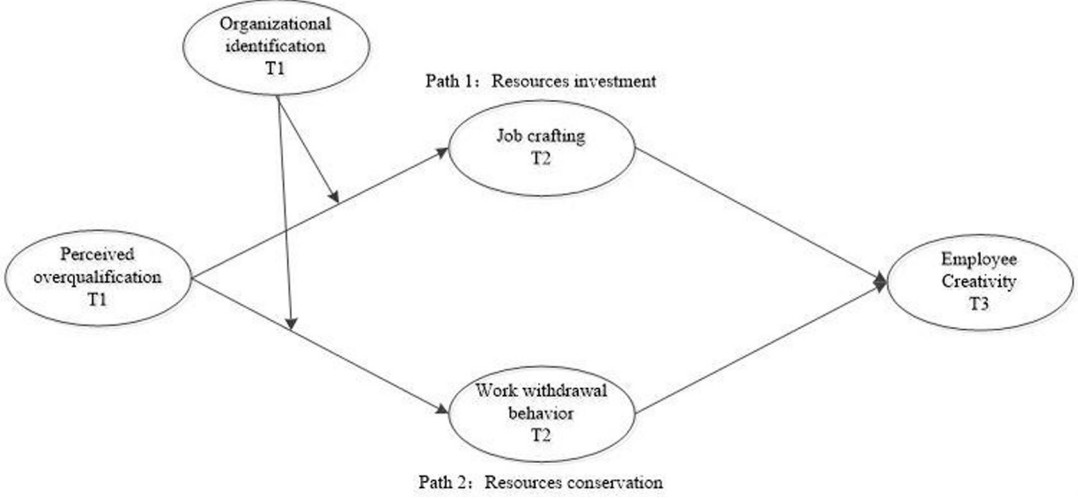

**Fig 1. Research model.**

perceived overqualification, organizational identification, and demographic variables (gender, age, education level, and tenure). Three weeks later, in Phase 2 (T2: June 2022), employees report job remodeling and work withdrawal behaviors. Three weeks later, in Phase 3 (T3: July 2022), employees report creativity. After matching based on the codes, questionnaires that were invalid, regular, or had too much missing data were removed, resulting in a valid sample of 487, with a valid sample response rate of 87.6%.

The sample features are as follows: 45.6% were male and 54.4% were female; The age of 25 and below accounted for 5.1%, the age of 26–35 accounted for 52.4%, the age of 36–45 accounted for 34.1%, and the age of 46 and above accounted for 8.4%; High school and below education accounted for 11.5%, junior college education accounted for 29.4%, undergraduate education accounted for 46.8%, and graduate education and above accounted for 12.3%; The proportion of working years in this position is 13.3% for 1 year or less, 33.9% for 1–5 years, 42.5% for 5–10 years, and 10.3% for 10 years or more.

## Measures

All of the measurement scales were established well and drawn from the literature. All items used the same seven-point Likert scale format ranging from 1 (strongly disagree) to 7 (strongly agree).

Perceived overqualification: Perceived overqualification was measured using Maynard et al.'s nine-item scale [2]. Sample items include "The level of education required for my job is lower than my current education." and "People with less education than me can do my current job well." etc. The Cronbach's alpha for the scale was 0.902.

Job crafting: Job crafting were measured using Leana et al.'s four-item scale [49]. Sample item include "I always introduce new ways of working to improve my work." etc. Cronbach's alpha for the scale was 0.868.

Work withdrawal behavior: Work withdrawal behavior measured using Lehman and Simpson's twelve-item scale [50]. Sample items include "I am distracted at work" and "I attend to personal matters during work hours", etc. Cronbach's alpha for the scale was 0.918.

Employee creativity: Employee Creativity was measured using Farmer's four-item scale. Sample item include "In the working process, I give priority to trying new ideas or new methods." etc. Cronbach's alpha for the scale was 0.862.

Organizational identification: Organizational identification was measured using Maeland Ashforth's six-item scale. Sample item include "When someone praises the organization, it feels like a personal compliment to me." etc. Cronbach's alpha for the scale was 0.856.

Control variables: gender, age, education level, and tenure were selected as control variables.

## Results

### Confirmatory factor analysis

Before testing the hypotheses, we conducted a confirmatory factor analysis (CFA) using AMOS 22.0 to test the discriminant validity of five latent variables: perceived overqualification, job crafting, work withdrawal behavior, employee creativity, and organizational identification. The results of CFA were presented in Table 1, the proposed model (i.e., the five-factor model) fitted better than alternative models ($\chi 2$ = 1084.581, df = 445, RMSEA = 0.054, IFI = 0.923, TLI = 0.913, CFI = 0.922), providing evidence for the distinctiveness of study variables.

**Table 1. Result of CFA.**

| Model | χ2 | df | χ2/df | RMSEA | IFI | TLI | CFI |
|---|---|---|---|---|---|---|---|
| [a] One-factor | 4196.018 | 455 | 9.222 | 0.130 | 0.547 | 0.504 | 0.545 |
| [b] Two-factor | 3126.264 | 454 | 6.886 | 0.110 | 0.677 | 0.645 | 0.675 |
| [c] Three-factor | 2732.693 | 452 | 6.046 | 0.102 | 0.724 | 0.696 | 0.723 |
| [d] Four-factor | 1361.382 | 449 | 3.032 | 0.065 | 0.890 | 0.877 | 0.889 |
| [e] Five-factor | 1084.581 | 445 | 2.437 | 0.054 | 0.923 | 0.913 | 0.922 |
| [f] Six-factor | 1079.277 | 444 | 2.431 | 0.054 | 0.923 | 0.914 | 0.923 |

N = 487. RMSEA = Root Mean Square Error of Approximation, POQ = Perceived overqualification, JC = Job crafting, WWB = Work withdrawal behavior,

EC = Employee Creativity, OI = Organizational identification.

[a] One-factor = all variables merged.

[b] Two-factor = POQ, JC+WWB+EC+OI.

[c] Three-factor = POQ, JC, WWB+EC+OI.

[d] Four-factor = POQ, JC, WWB, EC+OI.

[e] Five-factor = POQ, JC, WWB, EC, OI.

[f] Six-factor = Five-factor, CMB.

## Evaluation of common method bias (CMB)

First, Harman Single-factor test method was used to test the common method bias among variables. The results showed that a total of five factors were parsed for all question items, explaining a total of 61.670% of the variance. The first principal component explained 25.831% of the total variation, which did not exceed half of the total variance explained, indicating that the common method bias of data was within the acceptable range. Second, according to Podsakoff et al. [51], this study used the unmeasured latent method construct (ULMC) to test the effect of common method variance. That is, the common method bias is used as a latent variable, and the other factors are made to load on this latent variable to construct a structural equation model for comparison with the five-factor model. According to Table 1, compared to the five-factor model, the six-factor model after adding the common method bias factor, the changes of RMSEA, IFI, TLI, and CFI were less than 0.002, and the changes were not significant, which further indicated that there was no significant common method bias in this study.

## Descriptive statistical analysis

The means, standard deviations, and correlation coefficients of the variables were shown in Table 2. The AVE (Average Variance Extracted) values in diagonal brackets showed that the AVE values of each variable were greater than 0.5, which indicated good convergent validity among the variables; the square root of AVE values of each variable were greater than the correlation coefficients with other variables, which also indicated good discriminant validity among the variables; the correlation coefficients among the variables were less than 0.7, which indicated that The correlation coefficients between variables are less than 0.7, indicating that there is no serious problem of multicollinearity in the study.

In addition, as shown in Table 2, perceived overqualification was significantly positively correlated with job crafting (b = 0.391, p<0.001), and job crafting was significantly positively correlated with employee creativity (b = 0.424, p<0.001), and the above correlations preliminary support for hypotheses H1-H3; perceived overqualification was significantly positively correlated with work withdrawal behavior(b = 0.324, p<0.001), and work withdrawal behavior was significantly negatively correlated with employee creativity (b = -0.414, p<0.001), and the above correlations preliminarily supported hypotheses H4-H6; organizational identification

**Table 2. Descriptive statistical analysis results.**

| | 1 | 2 | 3 | 4 | 5 | 6 | 7 | 8 | 9 |
|---|---|---|---|---|---|---|---|---|---|
| 1 Gen | 1 | | | | | | | | |
| 2 Age | 0.084 | 1 | | | | | | | |
| 3 Edu | 0.044 | 0.078 | 1 | | | | | | |
| 4 Ten | 0.040 | 0.219*** | -0.009 | 1 | | | | | |
| 5 POQ | -0.075 | 0.023 | 0.135** | 0.047 | 1(0.579) | | | | |
| 6 JC | -0.040 | 0.064 | 0.086 | 0.065 | 0.391*** | 1(0.630) | | | |
| 7 WWB | 0.015 | 0.030 | 0.085 | 0.005 | 0.324*** | -0.132** | 1(0.582) | | |
| 8 EC | -0.040 | 0.030 | -0.050 | 0.068 | 0.069 | 0.424*** | -0.414*** | 1(0.558) | |
| 9 OI | -0.026 | 0.010 | 0.055 | -0.055 | 0.143** | 0.187*** | -0.103* | 0.063 | 1(0.521) |
| M | 1.544 | 2.458 | 2.600 | 2.497 | 4.613 | 5.113 | 4.562 | 4.871 | 4.932 |
| SD | 0.498 | 0.721 | 0.847 | 0.851 | 1.054 | 0.895 | 1.137 | 0.895 | 0.742 |

*, **, and ***, respectively, indicate significance at the level of p<0.05, p<0.01, and p<0.001. POQ = Perceived overqualification, JC = Job crafting, WWB = Work withdrawal behavior, EC = Employee Creativity, OI = Organizational identification.

was significantly positively correlated with job crafting (b = 0.187, p<0.001) and significantly negatively correlated with work withdrawal behavior(b = -0.103, p<0.05), and preliminarily supported hypotheses H7-H10.

## Hypothesis testing

**Testing main effects and mediating effects.** This study conducted a path analysis using Mplus 8.3. The results were shown in Fig 2. After controlling the control variables such as gender, age, education level and tenure, perceived overqualification had a significant positive effect on job crafting (b = 0.331, p<0.001), and job crafting had a significant positive effect on employee creativity (b = 0.349, p<0.001). Hypothesis H1 and H2 were supported; Perceived overqualification had a significant positive effect on work withdrawal behavior (b = 0.358, p<0.001), and work withdrawal behavior had a significant negative effect on employee creativity (b = -0.304, p<0.001). Hypothesis H5 and H6 were supported.

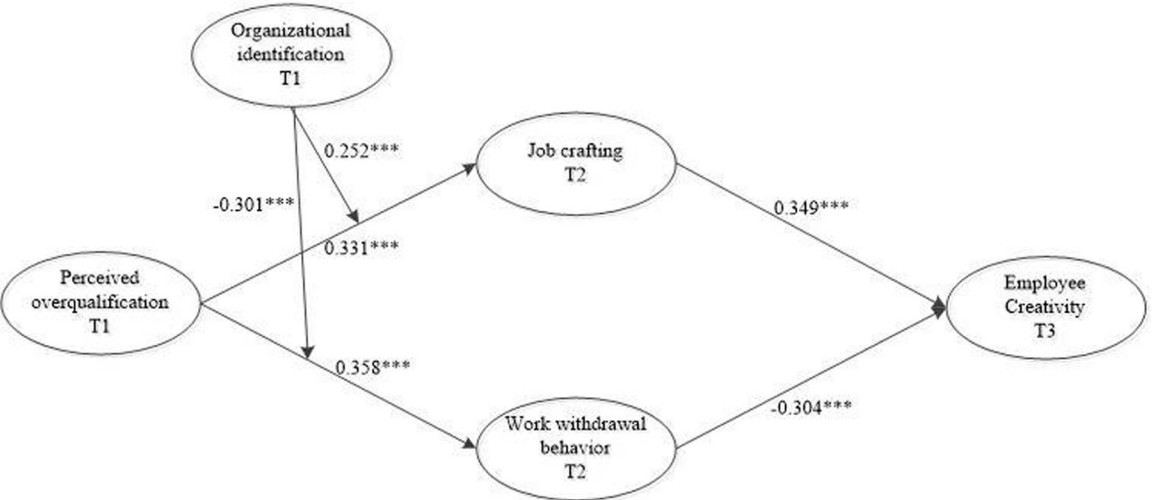

**Fig 2. Results of the theoretical model.** *** *indicate significance at p<0.001.*

This study uses bootstrap method to test the mediating effect. The results were shown in Table 3. According to Table 3, the indirect effect of perceived overqualification on employee creativity through job crafting was 0.116, with a 95%CI of [0.077,0.158], excluding 0; The indirect effect of perceived overqualification on employee creativity through work withdrawal behavior is -0.109,with a 95%CI of [-0.161, -0.066], excluding 0. Therefore, hypothesis H3 and H6 were supported.

**Testing moderating effects.** This study used the latent moderated structural equations [52] to test the moderating effect of organizational identification, and the results were shown in Fig 2. the interaction term of perceived overqualification and organizational identification had significant effects on job crafting (b = 0.252, p<0.001) and work withdrawal behavior(b = -0301, p<0.001), indicating that organizational identification moderated the relationship between perceived overqualification and job crafting, and perceived overqualification and work withdrawal behavior, respectively.

To further explain the moderating effect of organizational identification, we conducted a simple slope test according to the method recommended by Aiken et al. [53]. As shown in Fig 3, the positive effect of perceived overqualification on job crafting was weaker when the level of organizational identification was low, and on the contrary, the positive effect of perceived overqualification on job crafting was enhanced when the level of organizational identification was highly significant. Therefore, H7 was supported. As shown in Fig 4, the positive effect of perceived overqualification on work withdrawal behavior was stronger when the level of organizational identification was low, and on the contrary, the positive effect of perceived overqualification on work withdrawal behavior was weaker when the level of organizational identification was high. Therefore, H8 was supported.

**Testing moderated mediation effects.** This paper used the "bootstrapping method" recommended by Edwards and Lambert to analyze the mediating effects of job crafting and work withdrawal behavior between perceived overqualification and employee creativity at different levels of organizational identification [54]. The results were shown in Table 4:

1. When the mediating variable was job crafting, the positive effect of perceived overqualification on job crafting was 0.533 with a 95%CI of [0.076,0.243] (excluding 0) at high level of organizational identification; at low level of organizational identification, the positive effect of perceived overqualification on job crafting was 0.159 with a 95%CI of [0.434,0.626] (excluding 0); and the difference value of the effect between high level organizational identification and low level organizational identification was 0.374, with a 95%CI of [0.278,0.414] (excluding 0); these indicated that the positive effect of perceived overqualification on job crafting in the process of changing from low level organizational identification to high level organizational identification was strengthened. In addition, the indirect effect of perceived overqualification on employee creativity through job crafting at high level of organizational identification was 0.187 with a 95%CI of [0.131,0.246] (excluding 0); the difference value of the indirect effect between high level organizational identification and low level

**Table 3. Bootstrapping test results of mediating effect.**

| Effect path | Mediate effect | 95% confidence interval | |
|---|---|---|---|
| | | Lower limits | Upper limits |
| POQ→JC→EC | 0.116[***] | 0.077 | 0.158 |
| POQ→WWB→EC | -0.109[***] | -0.161 | -0.066 |

[***] indicate significance at p<0.001

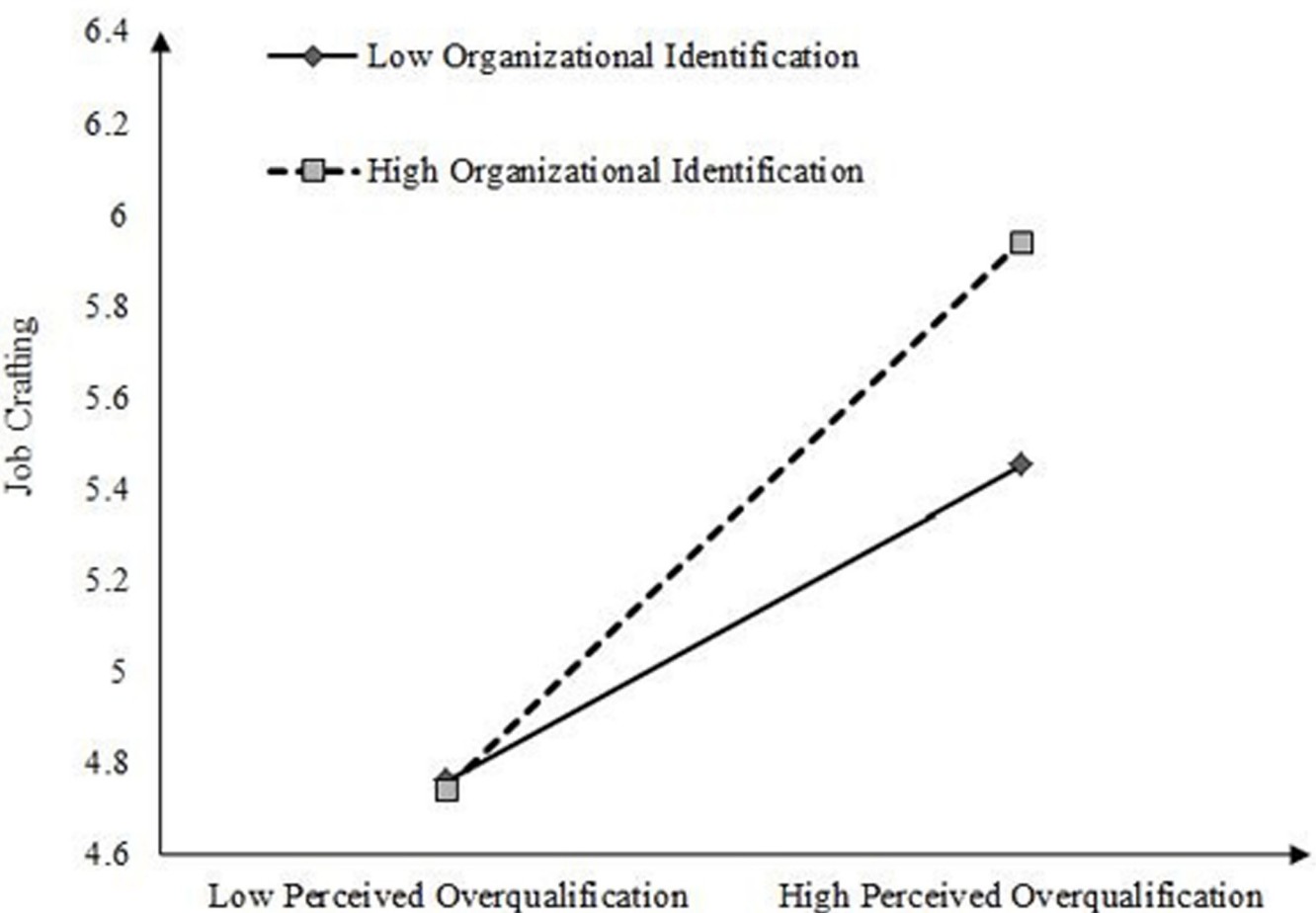

**Fig 3. The moderating role of organizational identification.**

organizational identification was 0.131, with a 95%CI of [0.064,0.156](excluding 0). Therefore, hypothesis H9 was supported.

2. When the mediating variable was work withdrawal behavior, the positive effect of perceived overqualification on work withdrawal behavior was 0.575 with a 95%CI of [0.465,0.685] (excluding 0) at a low level of organizational identification; the positive effect of perceived overqualification on work withdrawal behavior was 0.128 with a high level of organizational identification, with a 95%CI of [0.006,0.651] (excluding 0); and the difference value of the effect between high level organizational identification and low level organizational identification is -0.447, with a 95%CI of [-0.441,-0.262] (excluding 0); These indicated that the process of changing from low level organizational identification to high level organizational identification, the positive effect of perceived overqualification on work withdrawal behavior was weakened; In addition, the indirect effect of perceived overqualification on employee creativity through work withdrawal behavior at low level of organizational identification was -0.174 with a 95%CI of [-0.235,-0.118] (excluding 0); the difference value of the indirect effect between high level of organizational identity and low level of organizational identification was 0.135 with a 95%CI of [0.064, 0.156] (excluding 0). Therefore, hypothesis H10 was supported.

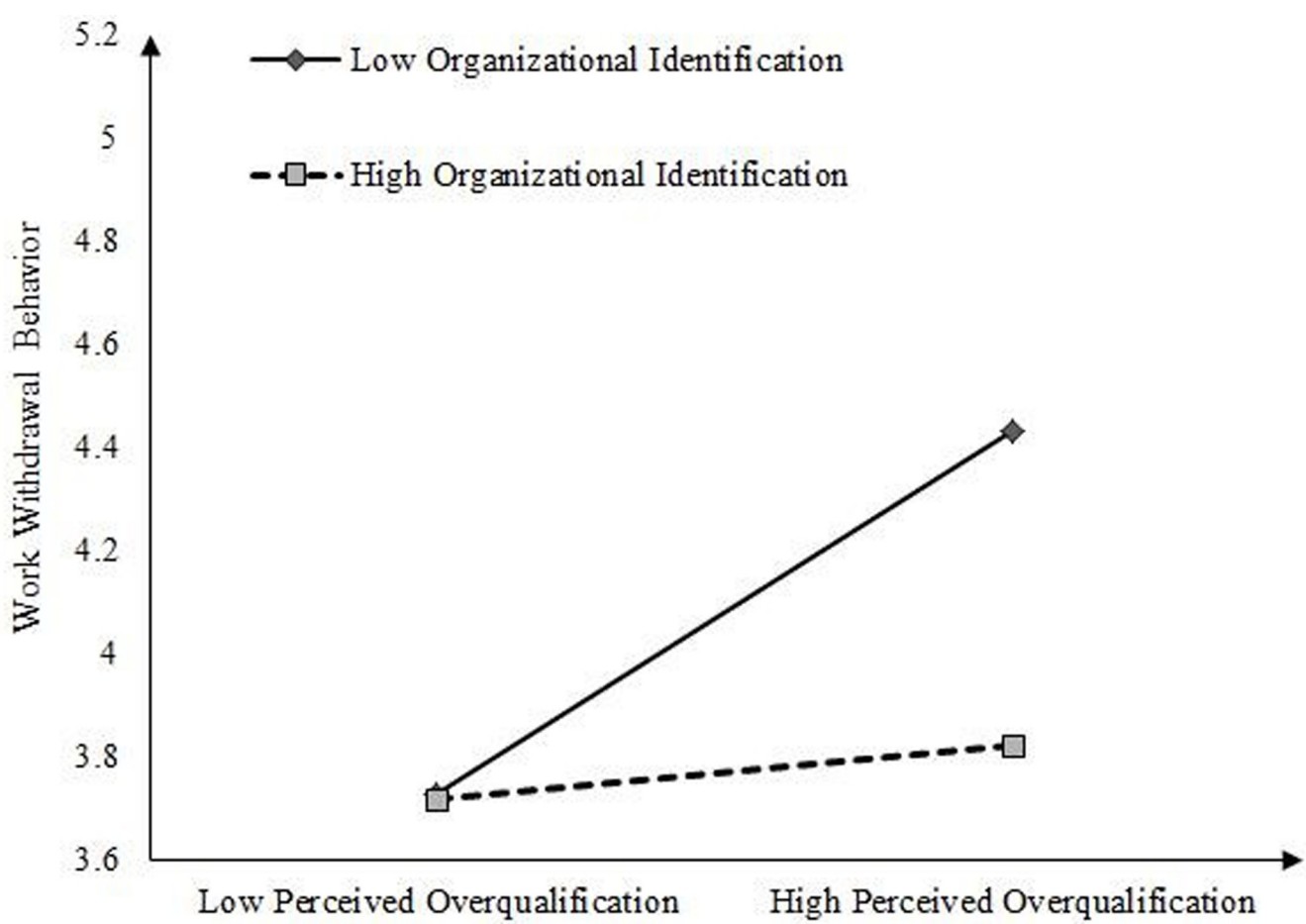

**Fig 4. The moderating role of organizational identification.**

## Conclusion and discussion

The purpose of this study is to investigate the mechanisms and boundary conditions of perceived overqualification on employees' creativity, and to improve the understanding of when and how perceived overqualification affects employees' creativity. The results suggest that perceived overqualification affects creativity both positively through job crafting and negatively through work withdrawal behavior. When employees feel they are overqualified, they may be able to take advantage of their ability to craft their work while completing it because they have the skills needed to perform at a high level, but they may also be prone to job dissatisfaction and frustration, which can lead to job withdrawal behaviors. In addition, the double-edged sword effect of overqualification is moderated by organizational identification, where high organizational identification can promote the positive effects of overqualification and weaken its negative effects.

## Theoretical implications

The first theoretical contribution of this study is the construction of a dual path model of the influence of perceived overqualification on employee creativity based on conservation of

**Table 4. Moderated mediation effect test.**

| Effect Path | B | SE | 95% confidence interval | |
|---|---|---|---|---|
| | | | Lower limits | Upper limits |
| **Moderating Effect** | | | | |
| POQ→JC(Mean + SD) | 0.533*** | 0.047 | 0.076 | 0.243 |
| POQ→JC(Mean - SD) | 0.159** | 0.043 | 0.434 | 0.626 |
| Difference | 0.374*** | 0.035 | 0.278 | 0.414 |
| POQ→WWB(Mean + SD) | 0.128* | 0.062 | 0.006 | 0.251 |
| POQ→WWB(Mean - SD) | 0.575*** | 0.056 | 0.465 | 0.685 |
| Difference | -0.447*** | 0.046 | -0.441 | -0.262 |
| **Moderated Mediation Effect** | | | | |
| POQ→JC→EC(Mean + SD) | 0.187*** | 0.029 | 0.131 | 0.246 |
| POQ→JC→EC(Mean - SD) | 0.056** | 0.022 | 0.015 | 0.103 |
| Difference | 0.131*** | 0.018 | 0.084 | 0.162 |
| POQ→WWB→EC(Mean + SD) | -0.039 | 0.027 | -0.096 | 0.013 |
| POQ→WWB→EC(Mean - SD) | -0.174*** | 0.030 | -0.235 | -0.118 |
| Difference | 0.135*** | 0.024 | 0.064 | 0.156 |

*, **, and ***, respectively, indicate significance at the level of p<0.05, p<0.01, and p<0.001.

resources theory. Given the differences in research perspectives and theoretical foundations, there is no consensus among academics as to whether perceived overqualification have a negative impact on employee creativity or not. Whether perceived overqualification promotes employee creativity is largely determined by the underlying mechanisms. The results of this study show that the effects of perceived overqualification on employee creativity are in two ways: "resources investment" and "resources conservation". In the resources investment path, perceived overqualification positively influences employee's creativity through job crafting; in the resource conservation path, perceived overqualification negatively influences employee's creativity through work withdrawal behavior. The effect direction of the two influence paths is different. In the resources investment path, the effect of perceived overqualification on employee creativity is positive. In the resources conservation path, the effect of perceived overqualification on employee creativity is negative. The above results not only answer the inconsistent findings of previous studies [10–13], but also contribute to a further comprehensive view of the effect of perceived overqualification on employees' work behavior and outcomes..

Second, conservation of resources theory is introduced into the research field of perceived overqualification, which expands the application scope of conservation of resource theory and enriches the theoretical perspective of the field of overqualification. So far, most studies on the sense of overqualification have been based on the relative deprivation theory [55], the perspectives of person-job matching theory [34], and the equity theory [9]. Very few studies based on the conservation of resources theory only focus on the negative effect of perceived overqualification caused by resource depletion [13]. This study comprehensively introduces conservation of resources theory, and the results show that the positive and negative effects of perceived overqualification coexist, responding to the call of scholars to comprehensively explore the effect mechanism of the sense of overqualification from the perspective of integration [56].

Finally, this study confirms the important moderating role of organizational identification in the influence mechanism of perceived overqualification on employee's creativity and further clarifies the boundary conditions of perceived overqualification on employee's creativity. As an important boundary condition of the resource investment path and resource conservation

path, organizational identification determines, to a certain extent, the attitude of overqualified employees to their surplus resources and their perception of work itself, which in turn affects their work behavior. By examining the moderating effect of organizational identification in the path of perceived overqualification on employees creativity, this study further clarifies the important role of organizations in managing high-quality talent and provides practical guidance for managing overqualified employees in organizations.

## Practical implications

Firstly, the organization should take an objective view of overqualification when selecting employees. The results of this study show that perceived overqualification can promote employees to improve their creativity by their job crafting behaviors, thus bringing positive effects; it will also stimulate employees' work withdrawal behavior, which will bring negative effects to organizations and individuals. Therefore, managers should take a reasonable view of overqualification. They should not only correctly identify and make good use of the potential value of overqualified employees to the organization, but also be vigilant about the possible negative effects of overqualified employees. Specifically, when selecting and hiring employees with obvious overqualification, organizations should further investigate their job hunting intention rather than directly reject them. For employees who are really willing to be "over-qualified", appropriate incentives and guidance can effectively promote them to show more positive behavior, use their personal qualifications beyond the job demands to exert their creativity, and help achieve organizational goals.

Secondly, it is very important to improve the motivation of overqualified employees. The mediating effects of job crafting and work withdrawal behavior indicate that for employees with perceived overqualification, job motivation rather than job competence is the key factor in shaping their positive behavior. Therefore, managers should focus less on assigning tasks to employees depending on their abilities and more on how to improve the motivation of highly qualified employees. On the one hand, training managers in HRM can apply a number of measurement tools to assess and track their perception of overqualification. For those who are identified as clearly overqualified, their direct supervisors can try to raise the job demands or performance standards to avoid work withdrawal behavior. On the other hand, providing opportunities or platforms for overqualified employees to encourage them to use their surplus resources to improve job crafting behavior, such as taking the initiative to point out problems in their work and correct them, to produce a series of positive results; establishing an appropriate pay-for-performance incentive system will have an extrinsic motivating effect on overqualified employees and motivate them to devote more personal resources to their work.

Finally, organizations should emphasize on developing employees' organizational identification. The findings suggest that organizational identification weakens the negative effects of perceived overqualification and strengthens the positive effects of perceived overqualification. When employees with a high level of perceived overqualification have a high level of organizational identification, they show more job crafting behaviors and less work withdrawal behaviors; in contrast, employees with a low level of organizational identification show more work withdrawal behaviors, which may harm organizational interests. Therefore, managers should focus on fostering employees' organizational identification. Specifically, when selecting and hiring new employees, managers should pay more attention to selecting employees who fit with the organization, such as adding value tests in the recruitment process. In addition, the organization can promote the organization's vision, mission and values by organizing activities such as group training to internalize the organization's values into employees' personal values, thus improving employees' sense of organizational identification.

## Limitations and directions

Despite its contributions, this study holds some limitations. First, although the data in this study are collected at three time points, there are still some limitations in proving the causal relationship of variables. Future research can further verify the causal relationship between variables by using situational experiments or empirical sampling methods (ESM). Second, the measures were all self-reported, which raises the possibility of common method bias. In addition, self-reporting is a data collection method that is easily affected by the subjective feelings of employees. In the future, the work performance of employees can be obtained by combining the supervisor's report with the personal report. Finally, this study takes job crafting as a whole construct, and does not distinguish the specific motivation of employees' job crafting behavior. The impact of job crafting with different motivations on work results is different. For example, Job crafting towards strengths are beneficial to the organization, while and job crafting towards interests is based on personal interests, which may not be beneficial to the organization. Future research can distinguish the motivation of employees' job crafting and explore the differences in the impact of different types of job crafting behaviors triggered by perceived overqualification on employees' creativity.

## Supporting information

**S1 Data.**
(XLS)

## Author Contributions

**Conceptualization:** Lei Ning.

**Formal analysis:** Lei Ning, Yiting Zhang.

**Investigation:** Daokui Jiang, Yiting Zhang.

**Methodology:** Daokui Jiang, Lei Ning.

**Writing – original draft:** Lei Ning, Yiting Zhang.

**Writing – review & editing:** Daokui Jiang.

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
