## [Decision Letter · Decision Letter 0]

1 Mar 2024

PONE-D-23-36186Perceived overqualification as a double-edged sword for employee creativity: The mediating role of job crafting and work withdrawal behaviorPLOS ONE

Dear Dr. Jiang,

Thank you for submitting your manuscript to PLOS ONE. After careful consideration, we feel that it has merit but does not fully meet PLOS ONE’s publication criteria as it currently stands. Therefore, we invite you to submit a revised version of the manuscript that addresses the points raised during the review process.

We look forward to receiving your revised manuscript.

Kind regards,

Larissa M. Batrancea

Academic Editor

PLOS ONE

Journal Requirements:

3. In the online submission form, you indicated that the datasets presented in this article are not readily available because the data contains personal information that cannot be easily anonymized. Requests to access the datasets should be directed to the corresponding author.

Reviewers' comments:

Reviewer's Responses to Questions

**Comments to the Author**

1. Is the manuscript technically sound, and do the data support the conclusions?

Reviewer #1: Yes

Reviewer #2: Yes

Reviewer #3: Yes

Reviewer #4: Yes

2. Has the statistical analysis been performed appropriately and rigorously? 

Reviewer #1: Yes

Reviewer #2: Yes

Reviewer #3: Yes

Reviewer #4: Yes

3. Have the authors made all data underlying the findings in their manuscript fully available?

Reviewer #1: Yes

Reviewer #2: Yes

Reviewer #3: Yes

Reviewer #4: Yes

4. Is the manuscript presented in an intelligible fashion and written in standard English?

Reviewer #1: Yes

Reviewer #2: Yes

Reviewer #3: Yes

Reviewer #4: Yes

5. Review Comments to the Author

Reviewer #1: Recommendations for Manuscript ID PONE-D-23-36186 „Perceived overqualification as a double-edged sword for employee creativity: The mediating role of job crafting and work withdrawal behavior” for the PLOS ONE Journal.

General Comments

From my point of view, it is a very interesting topic and simultaneously it seems that to the best of my knowledge is the first empirical study reveals the micro mechanism and boundary conditions of the influence of excessive qualification on employee creativity. This study analyzed 487 valid samples obtained in three stages. The results show that: (1) Job crafting has a positive mediating effect on perceived overqualification and creativity, and the path of the two halves is positive; (2) Work withdrawal behavior plays a negative mediating role between the perceived overqualification and creativity. The path in the first half is positive, and the path in the second half is negative; (3) Organizational identity moderates the effect of perceived overqualification on job crafting and work withdrawal behavior. Specifically, the higher the sense of organizational identification, the stronger the positive effect of perceived overqualification on job crafting and the weaker the positive effect on work withdrawal behavior; (4) Organizational identification moderates the mediating role of job crafting and work withdrawal behavior in the relationship between overqualification and creativity. Specifically, the higher the organizational identity, the stronger the indirect positive effect of perceived overqualification on creativity through job crafting, and the weaker the indirect negative impact of perceived overqualification on creativity through work withdrawal behavior. The study conclusion deepens the research on the mechanism of the influence of the perceived overqualification on employees' work behavior, and provides practical enlightenment for the organization and management of employees with excess qualification.

The paper contains the following sections: Introduction, Theoretical background and hypotheses, Method, Results, Conclusion and Discussion, Theoretical Implications, Practical Implications, Limitations and Directions..

However, I find some recommendations:

1. I suggest to the authors that the last section Conclusions and Policy Implications.

2. The abstract must contain the main purpose of the paper, the research method used in the research and the main contributions.

3. It would be very useful to add the "Introduction" section and the purpose, objectives and hypothesis of the research. I consider that a weak point of the paper is that the authors did not show the novelty of the paper compared to other works. That is why, I consider that the introduction should specify the novelty of the paper compared to other papers published in this area.

4. The research is well based on science and the results are in agreement with the theoretical part. From my point of view, the paper is original and the topic addressed brings added value to the specialized literature regarding economic growth. The paper is well written and easy to read.

5. The research is well based on science and the results are in agreement with the theoretical part. The model applied to the analyzed data is correctly used in the analysis undertaken, it is a strength point of this paper.

6. Authors must specify the software used (STATA, Eviews, SPSS, etc.).

7. It is very important for the authors to analyze the descriptive analysis (with Kurtosis test, Jarque Berra test and interpretation, Skewness and Kurtosis interpretation). In the same time the correlation analysis and the VIF test are very important in this research.

11. I recommend the authors to refer to other recent works indexed in Web of Science, because only some cited works is not enough for a scientific paper. In my opinion, the authors must cite other papers regarding this subject and other subjects such as: tax compliance, economic growth etc. That is why, I suggest that the authors cite papers published in Web of Science Journals, such as:

1. Batrancea,L.M, Kudła,J., Błaszczak,B., Kopyt, M. (2022) Differences in tax evasion attitudes between students and entrepreneurs under the slippery slope framework, Journal of Economic Behavior & Organization, Volume 200,Pages 464-482,ISSN 0167-2681,https://doi.org/10.1016/j.jebo.2022.06.017.

2. Batrancea, L.M. Determinants of Economic Growth across the European Union: A Panel Data Analysis on Small and Medium Enterprises. Sustainability 2022, 14, 4797. https://doi.org/10.3390/su14084797

3. Batrancea, L.M.; Balcı, M.A.; Chermezan, L.; Akgüller, Ö.; Masca, E.S.; Gaban, L. Sources of SMEs Financing and Their Impact on Economic Growth across the European Union: Insights from a Panel Data Study Spanning Sixteen Years. Sustainability 2022, 14, 15318. https://doi.org/10.3390/su142215318

4. Batrancea, L.M.; Balcı, M.A.; Akgüller, Ö.; Gaban, L. What Drives Economic Growth across European Countries? A Multimodal Approach. Mathematics 2022, 10, 3660. https://doi.org/10.3390/math10193660

5. Paul-Olivier Klein, Laurent Weill, Bank profitability and economic growth, The Quarterly Review of Economics and Finance, Volume 84, 2022, Pages 183-199, ISSN 1062-9769, https://doi.org/10.1016/j.qref.2022.01.009.

6. Batrancea, L.; Pop, M.C.; Rathnaswamy, M.M.; Batrancea, I.; Rus, M.-I. An Empirical Investigation on the Transition Process toward a Green Economy. Sustainability 2021, 13, 13151. https://doi.org/10.3390/su132313151.

7. Alexandre Schwinden Garcia, Roberto Meurer, Effects of a development bank on the profitability of commercial banks: Evidence for Brazil, The Quarterly Review of Economics and Finance, Volume 85, 2022, Pages 246-259, ISSN 1062-9769, https://doi.org/10.1016/j.qref.2022.03.008.

8. Batrancea L.M., Nichita R.A., Batrancea I. (2012), Tax Non-Compliance Behavior in the Light of Tax Law Complexity and the Relationship between Authorities and Taxpayers, Scientific Annals of the „Alexandru Ioan Cuza” University of Iaşi, Economic Sciences Section, vol. 59, nr. 1, 97–106

9. Nichita R.A., Bătrâncea L.M., (2012), The Implications of Tax Morale on Tax Compliance Behavior, Annals of University of Oradea: Economic Science, Tom XXI, nr. 1, 739–744.

10. Batrancea, L. (2021). "Empirical Evidence Regarding the Impact of Economic Growth and Inflation on Economic Sentiment and Household Consumption" Journal of Risk and Financial Management 14, no. 7: 336. https://doi.org/10.3390/jrfm14070336,ISSN:1911-8066

11. Batrancea LM, Nichita A, Balcı MA, Akgüller Ö (2023) Empirical investigation on how wellbeing-related infrastructure shapes economic growth: Evidence from the European Union regions. PLoS ONE 18(4): e0283277. https://doi.org/10.1371/journal.pone.0283277, ISSN:1932-6203

12. Pegkas, P., Staikouras, C., & Tsamadias, C. (2019). Does research and development expenditure impact innovation? Evidence from the European Union countries. Journal of Policy Modeling, 41, 1005−1025.

13. Batrancea, L.M. (2022) Determinants of Economic Growth across the European Union: A Panel Data Analysis on Small and Medium Enterprises. Sustainability 2022, 14(8), 4797. https://doi.org/10.3390/su14084797.

14.

15. Batrancea, L.M., Kudła, J., Błaszczak, B., Kopyt, M. (2023) A dataset on declared tax evasion attitudes of students and entrepreneurs from Poland under the slippery slope framework, Data in Brief,109183, ISSN 2352-3409, https://doi.org/10.1016/j.dib.2023.109183.

16. Batrancea, L. M., Nichita, A., De Agostini, R., Batista Narcizo, F., Forte, D., Mamede, S. D. P. N., Roux‐Cesar, A. M., Nedev, B., Vitek, L., Pántya, J., Salamzadeh, A., Nduka, E. K., Kudła, J., Kopyt, M., Pacheco, L., Maldonado, I., Isaga, N., Benk, S., & Budak, T. (2022). A self-employed taxpayer experimental study on trust, power, and tax compliance in eleven countries. Financial Innovation, 8(1), 1-23. [96]. https://doi.org/10.1186/s40854-022-00404-y, ISSN:2199-4730.

In conclusion, the article should be improve. It should also be enhanced with a review of the literature adequate to the subject and a broader interpretation and commentary of the research results.

Reviewer #2: Dear Authors,

Please review your abstract and provide information in regard to your objectives and research methods used

The literature support for your first three hypotheses is not sufficient; similar for H6.

In regard to your methodology, it should include the following information: (1) participants and procedure; (2) the measures that you used; (3) the analysis strategy; (4) the results where you need to enclose descriptive statistics and hypotheses testing. Please add the missing information.

What are the results of your Pilot study?

What software have you been using.

As for your Results chapter, please add literature accepted thresholds for all the measures used.

Please add a summary in regard to your Hypotheses results and see to it in the light of previous research.

Best regards,

Reviewer #3: As stated and in the minor revisions suggested edits, there are minimal issues and the data written is self explanitory. The author understands the information that is being presented to the community; therefore, I offered suggested edits within the first portion of the submission that is being proposed.

Reviewer #4: In the manuscript presented an innovative and interesting topic. The advantage of manuscript is a properly developed research part and an interesting topic.

The results of the study relate to previous research in this field.

The manuscript's advantage is also the inclusion of practical implications and a correct and up-to-date overview of references.

6. PLOS authors have the option to publish the peer review history of their article (what does this mean?). If published, this will include your full peer review and any attached files.

Reviewer #1: No

Reviewer #2: No

Reviewer #3: **Yes: **Dr. David James Wallace

Reviewer #4: No

---

## [Author Response · Author response to Decision Letter 0]

1 May 2024

Response:

Thank for your comment. Based on your suggestion, we have reorganized the abstract section. The main purpose of the paper is "In this study, we attempt to clarify the relationship between perceived overqualification and employee creativity from an integrated perspective, exploring both positive and negative pathways. " The research method used in the research is "Based on 487 valid samples collected in three stages, empirical analysis is conducted". And the main contributions contains "This study provides a theoretical explanation for the inconsistent results of previous studies on the relationship between perceived overqualification and employee creativity from a resources perspective; The study conclusion deepens the research on the mechanism of the influence of the perceived overqualification on employees' work behavior, and provides practical enlightenment for the organization and management of employees with excess qualification. "

Introduction: It would be very useful to add the "Introduction" section and the purpose, objectives and hypothesis of the research. I consider that a weak point of the paper is that the authors did not show the novelty of the paper compared to other works. That is why, I consider that the introduction should specify the novelty of the paper compared to other papers published in this area.

Response: 

Thanks for your comment. Based on your suggestion, we have reorganized the introduction section. We tired our best to create tension. In the first paragraph of the INTRODUCTION in the new manuscript, we have reviewed the current research of perceived overqualification and presented research gap. In the second paragraph of the INTRODUCTION in the new manuscript, we explain the reason for looking at the impact of perceived overqualification on creativity. In the third paragraph of the INTRODUCTION in the new manuscript, we explain the reason for choosing job crafting and work withdrawal behavior as mediating variables. And then, in the fourth paragraph of the INTRODUCTION part in the new version of the manuscript, we added the explanation of choose organizational identification as a moderating variable. We introduced the new set of mediators, the outcome and the moderator in INTRODUCTION.

Theoretical Background and Hypotheses: The literature support for your first three hypotheses is not sufficient; similar for H6.

Response:

Thanks for your comment. Based on your suggestion, we supplemented H1-H3 and H6 to make up for the lack of references, such as: "Individuals with more resources have more opportunities to invest in resources, forming a resource gain spiral. Therefore，overqualified employees have more opportunities to use their surplus skills to proactively reinvent their jobs to create a better work environment (Wu et al., 2015), thus reducing the difference between reality and their ideal job."

Method: In regard to your methodology, it should include the following information: (1) participants and procedure; (2) the measures that you used; (3) the analysis strategy; (4) the results where you need to enclose descriptive statistics and hypotheses testing. Please add the missing information.

Response:

Thanks for your comment. Based on your suggestion, we have reorganized the method section. 

(1) The participants and procedure of the paper is "The research was conducted in five companies in eastern China, and the process was as follows: First, after seeking permission from company leaders, the researchers contacted the department heads to introduce the purpose of the study and explain the potential benefits of the study……" (2) The measures of the paper are "All of the measurement scales were established well and drawn from the literature. All items used the same seven-point Likert scale format ranging from 1 (strongly disagree) to 7 (strongly agree).……"

(3) The analysis strategy of the paper is "Firstly, this study conducted confirmatory factor analysis and common method bias test using Amos 22.0.…… "

(4) The results of the paper please see the results section, including CFA、CMB、Descriptive Statistical Analysis and Hypothesis Testing.

Result: 

1. Authors must specify the software used (STATA, Eviews, SPSS, etc.).

Response:

Thanks for your comment. In the METHOD part, we have added the Analytical Strategy. In this section, we have provided a detailed explanation of the software we were using (Amos22.0, SPSS22.0, Mplus8.3).

2. It is very important for the authors to analyze the descriptive analysis (with Kurtosis test, Jarque Berra test and interpretation, Skewness and Kurtosis interpretation). In the same time the correlation analysis and the VIF test are very important in this research.

Response:

Thanks for your comment. Based on your suggestion, we have added the descriptive analysis in the method section. "In addition, as shown in Table 3, according to the skewness values, each variable is less than 0, and the distribution is negative skewness, with the tail on the left and the peak tip on the right. In terms of the kurtosis values, since all the variables were lower the threshold of 3……"

3. What are the results of your Pilot study?

Response:

Thank you for your question. In this study, we did not conduct a pilot study. We agree that it is effective to adopt a pilot study. In future studies, we will conduct a pilot study before the formal study.

4. As for your Results chapter, please add literature accepted thresholds for all the measures used.

Response:

Thanks for your comment. In response to your feedback, we have made the following revisions in the results section:

Marked acceptable thresholds for each indicator in CFA, such as: "If RMSEA is less than 0.06 and IFI, TLI, and CFI are greater than 0.9, it indicates that the model has a high degree of fit (Hair)."

Conclusion and Discussion:

1. Please add a summary in regard to your Hypotheses results and see to it in the light of previous research.

Response:

Thanks for your comment. We have added the summary in regard to our Hypotheses results, please see "The results show that: First, the "double-edged sword" effect of perceived overqualification has been verified. The impact of the sense of overqualification on creativity includes two ways……"

2. I suggest to the authors that the last section Conclusions and Policy Implications.

Response:

Thanks for your comment. We have adjusted the last section to Conclusions and Policy Implications.

References: I recommend the authors to refer to other recent works indexed in Web of Science, because only some cited works is not enough for a scientific paper. In my opinion, the authors must cite other papers regarding this subject and other subjects such as: tax compliance, economic growth etc.

Response:

Thanks for your comment. We have added a number of articles in this and other subjects, including those you recommended.

Thanks again for the valuable opinions of the experts on this article; I wish you good health and success in your work!

---

## [Editor Report · Decision Letter 1]

14 May 2024

Perceived overqualification as a double-edged sword for employee creativity: The mediating role of job crafting and work withdrawal behavior

PONE-D-23-36186R1

Dear Dr. Jiang,

We’re pleased to inform you that your manuscript has been judged scientifically suitable for publication and will be formally accepted for publication once it meets all outstanding technical requirements.

Kind regards,

Larissa M. Batrancea

Academic Editor

PLOS ONE

Additional Editor Comments (optional):

Congratulations!

Your paper has been accepted for publication.
---

## [Editor Report · Acceptance letter]

20 May 2024

PONE-D-23-36186R1 

PLOS ONE

Dear Dr. Jiang, 

I'm pleased to inform you that your manuscript has been deemed suitable for publication in PLOS ONE. Congratulations! Your manuscript is now being handed over to our production team.

Kind regards, 

on behalf of

Dr. Larissa M. Batrancea 

Academic Editor

PLOS ONE